# Impacts of Climate Change in the El Vizcaino Biosphere Reserve (REBIVI): Challenges for Coastal Communities and the Conservation of Biodiversity

**Antonina Ivanova Boncheva [1],\* and Pablo Hernández-Morales [2]**

1  Economics Department, Universidad Autonoma de Baja California Sur, La Paz 23080, Mexico
2  Marine and Coastal Sciences Department, Universidad Autonoma de Baja California Sur, La Paz 23080, Mexico
\*  Correspondence: aivanova@uabcs.mx; Tel.: +52-6121400442

**Abstract:** The impacts of climate change put pressure onto environmental and natural resources, which in turn increases the vulnerability of ecosystems and human communities. This makes social participation essential for biodiversity conservation. This article addresses the community perceptions of the adaptations made to climate change in the El Vizcaino Biosphere Reserve (REBIVI), in Baja California Sur (Mexico), one of the largest natural protected areas in Latin America. Workshops with local communities defined the Socio-ecological Assets for Conservation (SEACs) and prioritized the adaptation strategies and actions by multicriteria analysis. The conclusions point out that a combination of ecosystem-based adaptation (EbA) and communities-based adaptation (CbA) is the most appropriate pathway to affront the adverse impacts of climate change. Thus, a socioecological approach for land and biodiversity use planning has to be implemented if both human welfare and conservation are to be effectively promoted. The process is necessary and very important, as is the participation of the local residents in the implementation and monitoring of the adaptation actions.

**Keywords:** climate change; communities-based adaptation (CbA); ecosystems-based adaptation (EbA); El Vizcaino Biosphere Reserve; multicriteria analysis; natural protected areas; participatory community workshops (PCWs); socio-ecological assets for conservation

## 1. Introduction

The El Vizcaino Biosphere Reserve (REBIVI) is the largest natural protected area in Mexico and one of the largest protected areas of the American Continent. The landscapes of the REBIVI present a unique scenic beauty. Examples of the area's valuable assets are the natural attractions, represented by endemic species such as the pronghorn peninsular (*Antilocapra americana peninsularis*) and migratory species such as the gray whale (*Eschrichtius robustus*), as well as the cultural attraction of the cave paintings in Sierra de San Francisco, which have become a World Heritage Site [1]. That is why it is very important to explore the impacts of climate change in the REBIVI, in order to sustain the privileged environmental conditions of the area [2,3].

The communities settled in these semi-arid zones have to survive generally on ecosystem services, the availability of water being the most important. The future does not look promising and it is necessary to develop adaptation policies and instruments in the face of future climatic problems [4,5]. That is why it is crucial to assess the Socio-Ecological Assets for Conservation (SEACs) that are important for the ways of life of the local communities and to define adaptation strategies and actions based on the combined approach of ecosystem-based adaptation (EbA) and communities-based adaptation (CbA) [6–9].

Many scenarios of climatic conditions in the REBIVI model notable future changes, according to climate change scenarios [10]. Thus, the effects may harm the natural balance of the region [3,11] and the communities settled within the reserve, causing social, economic

and environmental conflicts [12]. The projections of the global climatic models suggest that the hydrological cycle will be affected in the quantity, intensity and distribution of the rain [13]. The average and maximum temperatures are expected to change, causing a greater frequency of droughts. The daily annual global precipitation is supposed to increase by more than 2% for the year 2080. However, some regions present a tendency for a fall in the average annual precipitation from 5 to 10% [13].

In addition to the predicted climate change impacts, the REBIVI is one of the most arid areas in the country. The local populations have already faced adversities due to this condition and they have adapted. Changes in the climate can cause serious negative consequences [14,15] and some have already been observed and suffered by the communities in the REBIVI (e.g., floods, droughts and biodiversity loss). However, this is also a reality in other regions of the world where habitat fragmentation, plant extinction, desertification and less available water, among other adverse effects, are well marked [16,17].

The perspective of the communities is also very important, since they contribute to detecting the favorable or adverse effects of climate change. Especially, the effects on the Socio-Ecological Assets of Conservation (SEACs) are highlighted, and the consequent negative influences on the communities' livelihood. All this contributes to a better evaluation, management, adaptability and resilience [3,18].

The objective of this paper is to define the adaptation actions to deal with the main vulnerabilities to climate change impacts in the REBIVI, based on the local inhabitants' perspectives, which were obtained in participatory community workshops (PCWs). As a result, an assessment was produced of the interrelation between the SEACs and the communities' welfare in the REBIVI, highlighting the climate change effects and the adaptation actions and projects for community resilience and socioecological change. Some key elements for the enabling environment were also defined, such as environmental education, training and access to technological innovation.

The Section 2 on methodology is divided into two parts. First, it describes the area of the study and climate change impacts on biodiversity and the communities, including the main productive activities. We also present the theoretical approaches to adaptation used in this paper: adaptation based on ecosystems (EbA) and adaptation based on communities (CbA). Second, we explain the methodology applied to the definition of the SEACs, and the prioritization by multicriteria analysis of adaptation actions in the PCWs. Section 3 is dedicated to outline the main results of the PCWs, presenting the adaptation actions by Strategic Adaptation Axes (SAAs), highlighting the benefits for the communities (emphasizing vulnerable groups), and the benefits for the SEACs. In the Section 4, we discuss the methodologically grounded proposals that strengthen adaptation practices. We present an overview of the actions by SAAs (water, agriculture, fisheries and tourism) outlining the direct dependence of the community welfare on the conservation of the natural resources and the adaptive climate action. Section 5 presents some concluding remarks and recommendations to advance the notion that a combination of ecosystem-based and community-based adaptations is the best socio-ecological pathway to contribute both to conservation and to community welfare.

## 2. Materials and Methods

### 2.1. Location and Climate

The REBIVI is located on the Baja California peninsula (Figure 1) and is part of the Sonoran Desert. Its climate is highly arid and presents arid characteristics, according to the Köppen climate classification. The area of the REBIVI is 2,546,700 hectares [19].

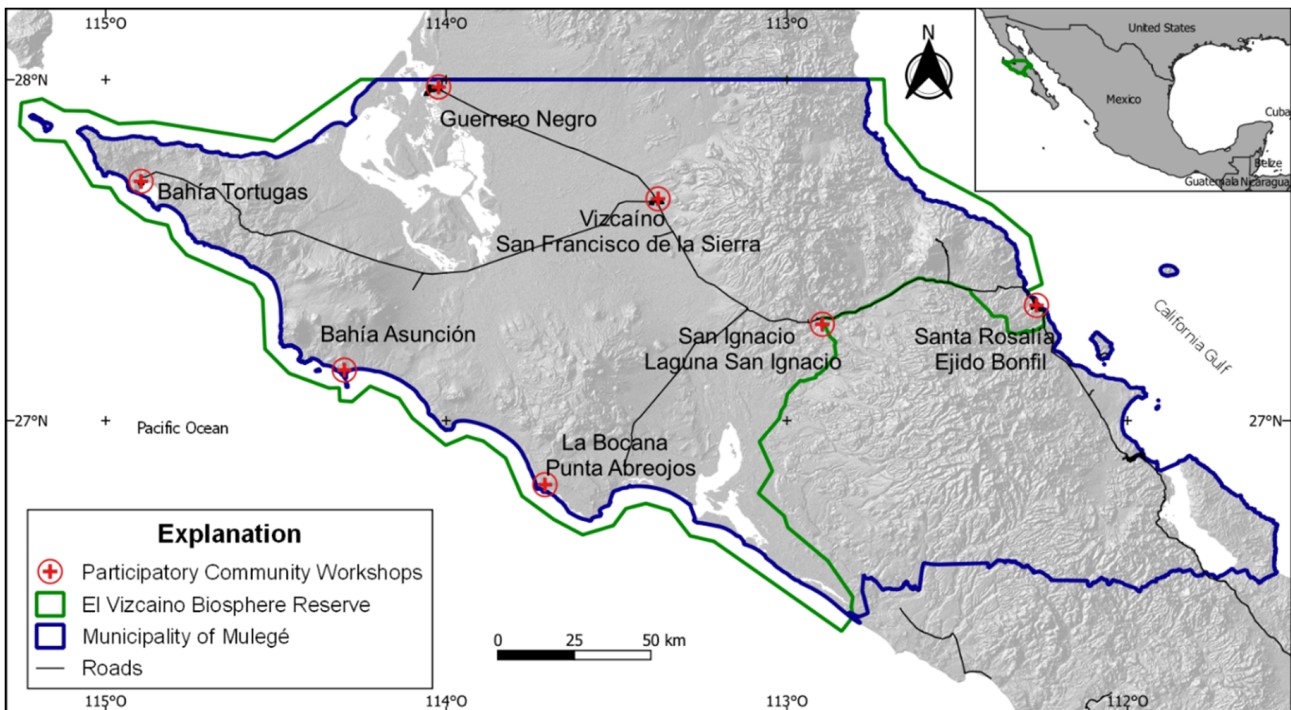

**Figure 1.** Location of the REBIVI and main settlements where the PCWs were carried out. Produced by authors based on digital data from [20,21].

Annual average temperatures are recorded between 18 °C–24 °C [22,23]. In the REBIVI, the largest area (approx. 80%) is defined as semi-warm, ranging from warm on the coast of the Gulf of California to the mountainous areas of San Francisco and Las Vírgenes (Figure 1). The average maximum temperature varies from 34 °C to 38 °C and the minimum varies from 5 °C to 10 °C [24].

The average annual precipitation in the REBIVI is calculated at 86.3–130.3 mm according to data from the Benito Juarez and El Mezquital climatic stations, respectively, with a record of 30 years from 1978 to 2011 [15]. The rain occurs mainly in summer (from May to October), with a lesser presence in winter (from November to April), although December is the month with the most precipitation on average. The average rainfall is between 50 and 70 mm per year [19,22]. In addition, the rainfall from tropical storms and hurricanes affects the REBIVI area during the Pacific cyclone season, mainly in summer [25].

The role of natural protected areas (NPAs), defined as geographical spaces which have managed to achieve the long-term conservation of nature with associated ecosystem services and cultural values [26,27], has become even more important in the context of global change [28,29]. Relevant to this are local inhabitants' economic practices and living standards since the lack of income alternatives, lifestyles and inequality are among the most common factors that constrain the implementation of adaptation and mitigation options and, in turn, strongly influence vulnerability to climate change and access and allocation issues [13,30].

Recognizing that improving community resilience to climate change in NPAs can initiate productive processes that both strengthen development opportunities and lead to improved biodiversity conservation [31,32].

The wide territory covered by the REBIVI and its low human population density make this protected area a valuable case study in terms of its potential for ecosystem conservation and enhancing the inhabitants' quality of life [18,33].

*2.2. Communities and Ecosystems: Adaptation Strategies and Projects*

Towards a Socioecological Change

Social participation is essential for biodiversity conservation and this is particularly clear in climate change conditions. The Intergovernmental Panel on Climate Change (IPCC) defines adaptation in human systems as the process of adjustment to actual or expected climate and its effects, in order to moderate harm or exploit beneficial opportunities [34]. Moreover, the IPCC underlines the argument that adaptation is imperative in order to address the impacts resulting from the warming due to past emissions. Thus, since adaptation plays an important role in conservation, an integrated ecosystem–community approach is needed to pursue effective conservation strategies [35,36].

Ecosystem-based adaptation (EbA) is the use of biodiversity and ecosystem services as part of an overall adaptation strategy to help people adapt to the adverse effects of climate change [33,35]. As one of the possible elements of an overall adaptation strategy, ecosystem-based adaptation uses the sustainable management, conservation and restoration of ecosystems to provide services that enable people to adapt to the impacts of climate change. It aims to maintain and increase resilience and reduce the vulnerability of ecosystems and humans to the adverse effects of climate change.

EbA can generate significant social, economic and cultural co-benefits, contribute to the conservation of biodiversity and build on the traditional knowledge and practices of indigenous peoples and local communities, including the important role of women as custodians of local knowledge. In addition, healthy, well-managed ecosystems have climate change mitigation potential, for example, through the sequestration and storage of carbon in healthy forests, wetlands and coastal ecosystems [5].

As an example of EbA actions that are already carried out in the REBIVI related to ecotourism, we can point out the Unities of Management for Wildlife Conservation (UMAs) for species in danger of extinction, the regulations for gray whale watching, the bans for species of flora and fishing commercial, game hunting, reforestation and land use planning of the reserve.

Community-based adaptation (CbA) has been defined as "a community-led process, based on communities' priorities, needs, knowledge and capacities, which should empower people to plan for and cope with the impacts of climate change" [12]. It refers to an evolving yet distinct set of principles and practices that consistently target the most vulnerable populations and focus on activities with the greatest direct impact [30]. This targeting and focusing, embedded in participatory situational analysis and action-planning processes, distinguishes it from the development of "business-as-usual" [35] approaches, which are often top-down and do not focus on the most vulnerable.

Adaptation strategies are generated through participatory processes that build on existing cultural norms and address the underlying causes of poverty that render some people especially vulnerable to the impacts of climate change. Influenced by this notion, the Global Environmental Facility Trust Fund and the United Nations Development Program financed climate change adaptation programs (CCAP) for NPAs, which are part of the Global Resilience Project. The project aims to reduce the direct and indirect adverse impacts of climate change on globally important biodiversity and human communities.

In Mexico, the project additionally sought to consolidate the institutional framework to implement the objectives of the Strengthening Management Effectiveness and Resilience of Protected Areas to Safeguard Biodiversity Threatened by Climate Change. In this context, it is important to increase the preparedness of NPA authorities and other stakeholders to address such a challenge [25].

*2.3. Participatory Community Workshops (PCWs)*

The main methodology tool was the participatory community workshops (PCWs) carried out in the REBIVI with local stake holders and community representatives during 2017–2018, which involved academic representatives, decision-makers, and 11 local communities to identify adaptation measures to reduce ecosystem and human vulnerability. In

first place, the SEACs were identified and then the adaptation actions were prioritized by multicriteria analysis.

The results of this paper are produced by the REBIVI's climate change adaptation program (CCAP-REBIVI) [3], financed during 2016–2018 by the United Nations Environment Program (UNEP) through Mexico's National Commission of Protected Natural Areas. Seven PCWs were held during 2017–2018 with representatives from eleven communities selected according to their population size, economic relevance and geographical location. The selected locations were Guerrero Negro, Vizcaíno, Villa Alberto Alvarado, San Francisco de la Sierra, San Ignacio, Laguna San Ignacio, Santa Rosalía, Ejido Bonfil, Punta Abreojos, Bahía Asunción and Bahía Tortugas (See Figure 1). Previous to the PCWs, the following main adaptation strategies were defined:

- Functional and healthy ecosystems;
- More resilient environment;
- Diversification of the ways of life.

Additionally, five Strategic Adaptation Axes (SAAs) were adopted according to the main productive activities in the REBIVI: (1) water, (2) fisheries and biodiversity, (3) agriculture and livestock, (4) tourism and (5) environmental education and research. Strategic Adaptation Axis I: water (SAA1W) was defined due to the importance of water resources as the basis for carrying out all productive activities, as well as maintaining the life of ecosystems and human beings. The SAA5 was highlighted as an important topic for the enabling environment for the implementation of the adaptation strategies and actions. Those Strategic Adaptation Axes had been previously defined in meetings of the research team with public officials of the REBIVI and the Natural Protected Area Commission (which is in charge of the Resilience Project in Mexico).

### 2.3.1. Definition of Socio-Ecological Assets for Conservation (SEACs)

The SEAC are a species or ecosystem that is vulnerable to the effects of climate change, or a species or ecosystem that provides important ecosystem services to human populations regardless of its vulnerability to climate change. Therefore its selection is justified by its importance within a global adaptive strategy (such is the case, for example, of the gray whale). To define the SEACs, we started from the classification of ecosystem services presented in a Millennium Ecosystem Assessment [37], as shown in the Table 1.

**Table 1.** Generic classification of the ecosystem services.

| Supporting Services (S) | Provisioning Services (P) | Regulating Services (R) | Cultural Services (C) |
|---|---|---|---|
| 1 Soil formation | 1 Food | 1 Air quality regulation | 1 Cultural diversity |
| 2 Photosynthesis | 2 Fiber | 2 Climate regulation | 2 Spiritual and religious values |
| 3 Primary production | 3 Fuel | 3 Water regulation | 3 Knowledge systems (traditional and formal) |
| 4 Nutrient cycling | 4 Genetic resources | 4 Erosion regulation | 4 Educational values |
| 5 Water cycling | 5 Biochemicals, natural medicines, and pharmaceuticals | 5 Water purification and waste treatment | 5 Inspiration |
| | 6 Ornamental resources | 6 Disease regulation | 6 Aesthetic values |
| | 7 Fresh water | 7 Pest regulation | 7 Social relations |
| | | 8 Pollination | 8 Sense of place |
| | | 9 Natural hazard regulation | 9 Cultural heritage values |
| | | | 10 Recreation and ecotourism |

Source: [37].

This table presents the ecosystem services in a generic way. However, Figure 2 (p. 8) shows which ecosystem service corresponds to each SEAC. The criteria to define the Socio-Ecological Assets for Conservation (SEACs) was based on the provision of ecosystem services for the main ways of life of the local communities.

2.3.2. Prioritization of Adaptation Actions: Multicriteria Analysis

A prioritization of the adaptation measures to climate change was performed according to a multicriteria analysis of the adaptation measures of the Strategic Adaptation Axes (SAAs) as a support tool in the decision-making process. Multicriteria analysis is used to facilitate decision making, so that different or even contradictory points of view may intervene. This is especially useful in planning because it allows for the integration of different criteria in a single framework of analysis, according to the intervention of the participants.

The objective of the PCWs was to gather the observations and comments of the REBIVI inhabitants to enrich and further specify the strategies and lines of action of each Strategic Adaptation Axis by social groups, main stakeholders and government agencies that would be responsible for their implementation and follow-up [38]. In addition, as part of the PCWs activities, the participants prioritized the actions of each Strategic Adaptation Axis based on a multicriteria analysis. Representatives from the municipal, state and federal levels of government, nongovernmental organizations, community members and producer associations (ranchers, fishermen and farmers) and educational institutions participated in this exercise.

The workshop attendees were organized into five working groups by SAAs, each with a facilitator (a member of the research team) and a rapporteur: (1) water, (2) agriculture, (3) fisheries and (4) tourism. The environmental education and research were also discussed as enabling factors.

After the facilitator of each thematic group presented the work dynamics, participants were asked to write each idea or piece of information on a sheet of paper, from which the facilitator synthesized the information (eliminating repetitions). This exercise resulted in the creation of adaptation measures for each strategic sector.

The discussion took place in two stages:

(a) Stage I. An assessment of climate change impact and the identification of adaptation measures.

At this stage, the impacts of climate change were identified by the assistants (e.g., drought, flooding, sea level rise, etc.). Then the level of each impact was ranked between 0 and 3 (0, no impact; 1, minimal impact; 2, considerable impact; and 3, high impact). Then the consequences of the impact were defined (e.g., drought affects crops, sea level rise affects the coastal infrastructure, etc.).

Next, participants defined the most appropriate adaptation strategies according to each impact. Once the adaptation strategies were formulated, there was discussion about whether or not any governmental institutions or civil society organizations received support for their implementation (e.g., financing, capacity building and environmental education).

(b) Stage II. The prioritization of the adaptation measures identified in stage I.

Multicriteria analysis is a decision-making support tool used during the planning process that allows the integration of different criteria according to the opinion of the actors in a single analysis framework to provide a comprehensive vision.

The prioritization was performed according to the multicriteria methodology [38]. The PCW participants discussed the following six criteria:

- Importance for the community;
- Contribution to conservation of the SEACs;
- Contribution to the vulnerable population;
- Resilience enforcement;
- Community participation;
- Institutional support (government and/or civil society organizations).

The participants ranked each of adaptation measures from 0 to 3 (0 does not contribute; 1 means a limited contribution; 2 a medium contribution; and 3 a strong contribution). Then, each group of answers was discussed. Subsequently, the ideas were debated until a consensus and possible solutions to the problem were identified. The results of these workshops are presented in the next section (see Figures S1 and S2, Supplementary Materials).

## 3. Results

*3.1. SEACs Identified in REBIVI*

The criteria to define the Socio-Ecological Assets for Conservation (SEACS) was based on the provision of ecosystem services for the main ways of life of the local communities, as already explained in the Section 2.

The workshops were designed to translate generic ecosystem services and their contribution to the communities' livelihoods into specific SEACs (see Table 2).

**Table 2.** SEACs identified in REBIVI.

| Marine Biodiversity | Terrestrial Biodiversity | Others |
|:---:|:---:|:---:|
| Gray Whale (SAA4T) | Shore Birds (SAA4T) | Streams/Springs (SAA1W & SAA2A) |
| Lobster (SAA3F) | Peninsular Pronghorn (SAA4T) | Wetlands (SAA1W y SAA3F) |
| Bivalve Molluscs (SAA3F) | Bighorn Sheep (SAA4T) | Islas–barrera (SAA3F) |
| Mangroves (SAA3F & SAA4T) | Cougar (SAA4T) | Salt Flats |
| Seagrasses (SAA3F & SAA4T) | Mule Deer (SAA4T) | |
| | Palm Groves (SAA1W & SAA4T) | |

Source: produced by the authors, based on the results of the PCWs.

It is important to note that, both in the research process and in the participatory community workshops, the inhabitants of the Reserve and the members of the research team highlighted the importance of some objects of a historical–cultural nature (Table 3). These are not strictly SEACs, but are important for the development of alternative tourism, which is an important way of life for the population, that fosters the resilience to the impacts of climate change. In addition, the residents of the REBIVI consider these assets

very important for their identity and cultural values. This process could be characterized as community innovation, which is important to include in adaptation strategies, as all these historical–cultural assets are exposed to the adverse climate effects.

Figure 2 lists the SEACs in the territory of the REBIVI, according to the ecosystem services (Table 1, p. 5) provided by each one of them. The ecosystem services concept has the potential to support communication and collaboration between actors in land use planning.

**Table 3.** Historical–Cultural Assets of REBIVI.

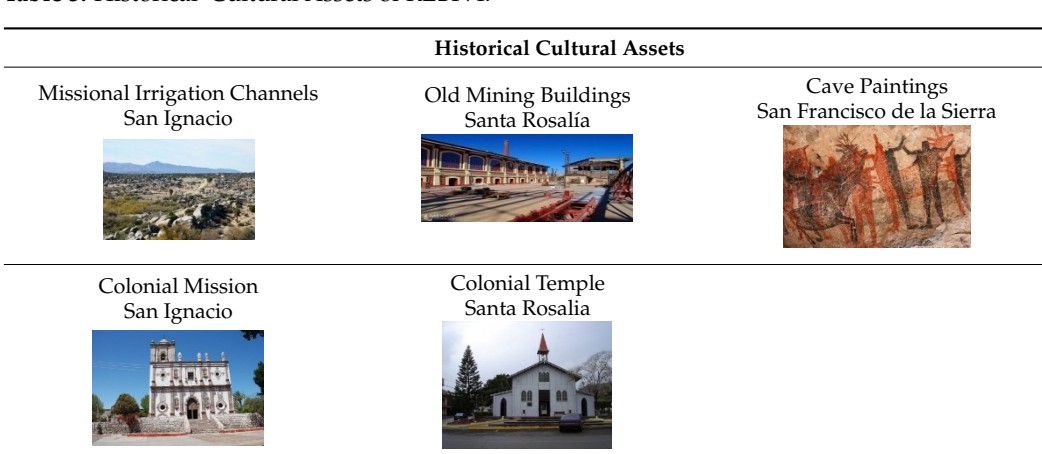

| Historical Cultural Assets | | |
|---|---|---|
| Missional Irrigation Channels San Ignacio | Old Mining Buildings Santa Rosalía | Cave Paintings San Francisco de la Sierra |
| Colonial Mission San Ignacio | Colonial Temple Santa Rosalia | |

Source: produced by the authors, based on the results of the PCWs.

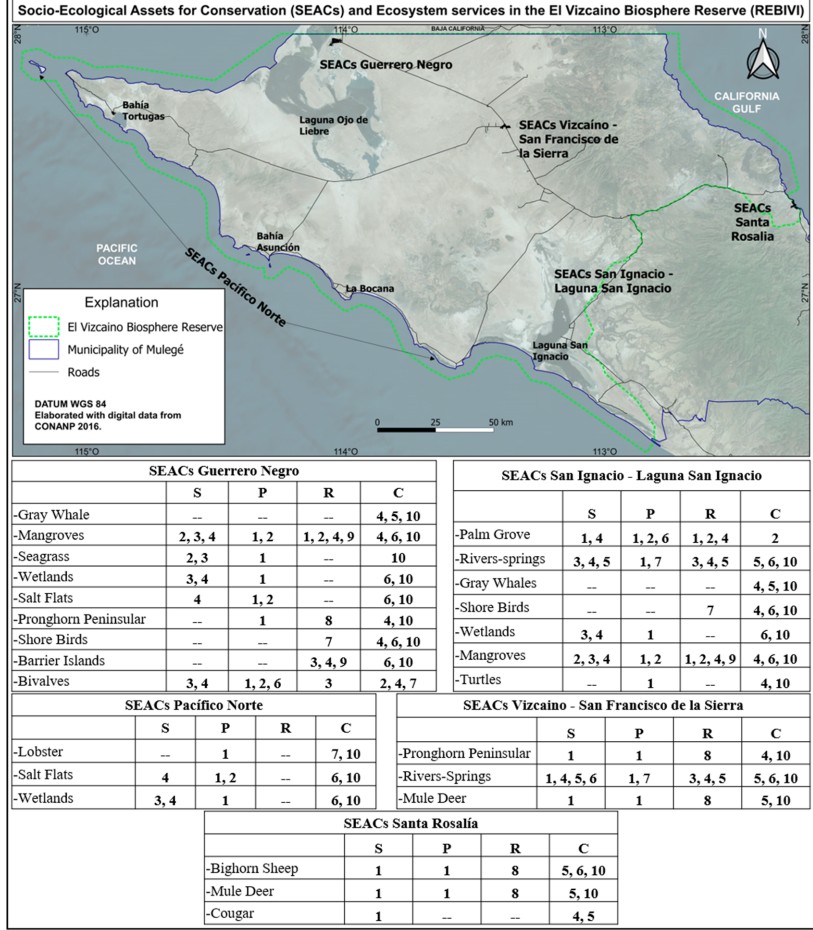

**Figure 2.** Socio-Ecological Assets for Conservation and Ecosystem Services in the El Vizcaino Biosphere Reserve according the generic classification (Table 1). Source: produced by the authors according to the results of the Participatory Community Workshops.

### 3.2. Adaptation Actions Defined and Prioritized in the PCWs by Strategic Adaptation Axes (SAAs)

The following tables present the adaptation measures, the beneficiary population, the beneficiary Socio-Ecological Assets for Conservation (SEACs) and the instances or organizations that could potentially be responsible for carrying out some of the measures.

Considering that planting in the Vizcaino area comes from the extraction of the aquifers, the lack of rain or the higher heat automatically implies a greater use of water. Thus, the elements of greatest impact were the appearance of pests and diseases, drought, increased solar radiation and frost. Interestingly, the presence of rains during the hurricane season (summer–fall) causes concern because they produce erosion and are a vector for diseases. One element to point out is that sprinkler irrigation is in place and workshop participants stressed the need to improve irrigation systems to make better use of water.

The technological advances refer to the drip irrigation and switch to water-saving crops. The participants in the workshops expressed their interest to be informed about the expected climate change effects and the possibilities that present the new technologies and the new forms of management to deal with these.

In the surrounding areas of the town of Vizcaino, extensive agro-export activity is developed. There is a marked recognition of the environmental problems that this type of agriculture represents in a desert region such as the REBIVI. Most of the families have their personal crops for family consumption, but they recognize the importance of integrating the communities' needs in hydrological studies. They also highlighted the importance of the effective regulation of sanitary conditions to prevent the introduction of pests that results from improper production practices.

In the exercise of defining and prioritizing the adaptation actions in the workshops, the most important issues to address were highlighted as the scarcity and poor quality of water, the appearance of pests, drought and extreme temperatures. In Vizcaino–San Francisco de la Sierra, technological improvement, training and climate information were prominent elements of adaptation action. However, greater emphasis was placed on the importance of better community organization and governance to promote the water saving and the access to financing.

A large number of the communities within the Reserve depend exclusively on fishing activity, which makes them vulnerable to changes in the availability of fishing resources. In this sense, the alternative activities proposed by the communities within the reserve were aquaculture in the first place, followed by eco-tourism and sport fishing. In the workshops, the need for studies on the feasibility of these alternative activities was raised. Additionally, the consensual planning for their best implementation was highlighted, considering that these do not contradict the regulations of the Reserve and have minimal interference with the productive activities already established.

The tourism sector throughout the El Vizcaino Biosphere Reserve is directly and indirectly linked to the global effects of climate change, specifically rising sea levels, floods, rising temperatures, droughts, water scarcity, floods and more extreme cyclones. The factors that are likely to affect the biodiversity of the REBIVI are the vulnerability to extreme events and water and food security, and therefore the human activities that are carried out there, including tourism practices [39].

Laguna Ojo de Liebre and Laguna San Ignacio, located in the REBIVI, are a breeding area for gray whales. Among gray whale populations in the world, this one remains the healthiest and most abundant [40] and can be closely observed and is friendly with humans (Figure S3, Supplementary Material). The whale watching is community managed in Laguna Ojo de Liebre and by small and medium enterprises in Laguna San Ignacio. This kind of ecotourism has contributed to raising the living standards of the local population and has promoted environmental protection and minimum impact practices [18].

Some further examples are the potential for the sustainable exploitation of palm trees (*Washingtonia robusta*) and the peninsular pronghorn game hunting. Since the REVIBI is a protected area, regulations prevent locals from harvesting palm tree leaves in ravine areas

that could be sold for thatching and could help prevent forest fires; thus, the reforestation programs must have a proper management plan as the oasis is an important tourism attraction. Regarding the priority species for conservation, the peninsular pronghorn was highlighted because it does not receive sufficient support in comparison with livestock and agriculture. The peninsular pronghorn is an emblematic example of resource management in the REBIVI, as the community has active participation in the game hunting of this species. This activity has contributed to increasing the population of the peninsular pronghorn (that was in danger of extinction before), and at the same time provided monetary income to the community.

The Tables 4–7 show the priorized adaptation actions for each Strategic Adaptation Axis (Table 4: SAA1 Water; Table 5: SAA2 Agriculture; Table 6: SAA3 Fisheries; Table 7: SAA 7: Tourism). As general result from the discussions, applying the combined EbA and CbA approach, complemented by special emphasis on vulnerable groups and gender issues, the REBIVI community members prioritized a combination of actions and projects, as follows:

- Foster livelihood resilience (productive activities diversification):
  - Develop ecotourism activities that could generate income for the community, while protecting biodiversity;
  - Develop aquaculture or seek new fishery resources;
  - Use hardier seed varieties and high-tech irrigation.

- Conserve or restore coastal wetlands, mangrove forests and/or woodlands;
- Take a holistic approach to watershed management;
- Employ "natural solutions" to reduce hazards (e.g., protecting or restoring mangrove to reduce the risk of sea level rise and coastal erosion);
- Minimize disaster risks and, thus, the impact of hazards, particularly on the most vulnerable groups and individuals;
- Strengthen local civil society and government institutions' capacities so that they can more effectively support communities and individual adaptation efforts.

Some more specific projects were also defined as a result of the workshops, and with the participation of experts:

- A study of extreme flood flows at local level;
- A feasibility study of rainwater harvesting systems in the mountains (dams);
- Implementing, updating and/or modernizing drinking water networks;
- Projects for protection structures against torrential floods;
- A study of the new uses and a defined approach to treated water;
- A feasibility analysis of desalination plants;
- A study of water availability and its quality in the Vizcaíno aquifer;
- The sea lions–coastal fishing conflict;
- The protection and reforestation of mangroves;
- Santa Rosalia Cetaceans: assess the abundance and diversity of cetaceans with potential for observation activities and ecotourism in the Santa Rosalia region;
- Assess the climate impacts on the gray whale abundance and its effects on ecotourism activity;
- Assess the impacts of climate variability to develop a predictive model of gray whale abundance that serves as a tool for strategic planning of whale watching activities;
- Large mammals of the Sierra: monitoring and a follow-up of the populations of large mammals (mainly puma, bighorn sheep and mule deer) as a tool for the development and follow-up of hunting activities and ecotourism;
- Using the peninsular pronghorn as a tool for the development and monitoring of ecotourism activity;
- The potential for bird watching to promote alternative tourism (birds from the oasis of San Ignacio and Birds from the coastal wetlands of Vizcaíno);
- The potential of cave paintings to promote tourism;

- A study of the potential for capturing and storing blue carbon in the various ecosystems and regions of the REBIVI (this project is oriented more to mitigation, but is closely related to ecosystems conservation).

Additionally to the specification of the adaptation strategies and projects, the integration of the community perspective was achieved, respecting their points of views and traditional knowledge, assuring their active participation in strategies for resolving the climate change adaptation-related issues and strengthening the smart governance [41]. The acceptance by local actors and their active participation in the design, implementation and monitoring of adaptation and remediation measures is crucial, strengthening resilience and reducing the impacts of climate change on natural resources and ecoservices and, consequently, improving the living conditions of the population.

**Table 4.** Prioritized adaptation actions of SAA1 "WATER" (multi-criteria analysis).

| Adaptation Actions | Benefits for Communities (Emphasizing Vulnerable Groups) | Benefits for SEACs |
|---|---|---|
| Organization and awareness on productive practices | Rancher inhabitants<br>Women: inclusion in higher remunerated activities. | Sheep Cimarron<br>Palm Groves |
| Infrastructure and technological advances | Benefits the rural population with the technical knowledge for the agriculture management in conditions of climate change<br>Women: I access to new technologies implies relief of physical efforts | Peninsular Pronghorn<br>Sheep Cimarron |
| Research and technical capacities building | Incudes all rural inhabitants | Conservation of ecosystems in general. |

Source: produced by the authors, based on the results of the PCWs.

**Table 5.** Prioritized adaptation actions of SAA2 "AGRICULTURE" (multi-criteria analysis).

| Adaptation Actions | Benefits for Communities (Emphasizing Vulnerable Groups) | Benefits for SEACs |
|---|---|---|
| Infrastructure and Technological advances | Benefits the rural population with the technical knowledge for the agriculture management<br>WomeIhe access to new technologies implies relief of physical efforts | Peninsular pronghorn<br>Sheep Cimarron |
| Research and technical capacities building | Includes all rural inhabitants | Conservation of ecosystems in general. |

Source: produced by the authors, based on the results of the PCWs.

**Table 6.** Prioritized adaptation actions of SAA3 "FISHERIES" (multi-criteria analysis).

| Adaptation Actions | Benefits for Communities (Emphasizing Vulnerable Groups) | Benefits for SEACS |
|---|---|---|
| Diversification of productive activities | Benefits all the members of communities (women, young people, children) | Lion's Claw<br>Abalone<br>Lobster<br>Scallops |
| Processing and marketing | Benefits all the members of communities (women, young people, children) | Wetlands<br>Lion's Claw<br>Abalone<br>Lobster<br>Scallops |
| Fisheries management | Benefits all the members of communities (women, young people, children) | Lion's Claw<br>Abalone<br>Lobster<br>Scallops |

Source: produced by the authors, based on the results of the PCWs.

**Table 7.** Prioritized adaptation actions of SAA4 "TOURISM" (multi-criteria analysis).

| Adaptation Actions | Benefits for Communities (Emphasizing Vulnerable Groups) | Benefits for SEACs |
|---|---|---|
| Prevention of landslides, floods and pollution in areas of tourist interest | Benefits women, children in places of risk (floods and landslides) | Springs<br>Palm Groves<br>Missional Irrigation Channels<br>Peninsular Pronghorn |
| Maintenance and improvement of sites of interest and access (Airport, road, dock) and adjacent areas; Infrastructure and renewables | The aIrway is a transfer option for the disabled, women and children;<br>The implementation of alternative energies replaces kerosene and firewood for cooking (benefits women and children) | Palm Groves<br>Mission San Ignacio<br>Colonial Church Santa Rosalia<br>Historical mining buildings Santa Rosalia<br>Cave paintings |
| Reforestation of palms and mangroves; regeneration of soil | Opportunity to create remunerated activities for women within the reforestation and seed collection. | Mangroves<br>Seagrasses<br>Palm groves<br>Peninsular Pronghorn<br>Mule Deer<br>Cougar |
| Ecotourism development | Benefit for the whole community<br>Opportunity to generate more productive activities for women. | Whales<br>Mangroves<br>Salt Flats<br>Seagrasses<br>Palm groves<br>Peninsular Pronghorn<br>Bighorn Sheep<br>Mule Deer<br>Cougar<br>Palm groves |

Source: produced by the authors, based on the results of the PCWs.

## 4. Discussion

In the workshops, a holistic approach was applied. It is vital to encourage and supervise public participation for a successful program; for example, the participants should not be restricted to the project executors, i.e., farmers or workers, but also students and scientists [42]. The participation was extended also to non-governmental organizations and small enterprises (mainly those involved with ecotourism).

In this discussion, we will analyze the main results by the Strategic Adaptation Axes. The participants in the workshops defined the Socio-Ecological Assets for Conservation. In general, a concern for wild flora and fauna was expressed. In Guerrero Negro, the attendees showed awareness about the existence of cave painting conservation initiatives (see Table 3). It is important to highlight the social innovation to define as SEACs some historical–cultural assets due, firstly, to their significance for the livelihoods of the inhabitants, and, secondly, to their vulnerability to the climate change impacts.

### 4.1. Water

The most alarming aspect of climate change in the area is the availability of water. The aquifers of the municipality present an overexploitation, because more water is extracted than is infiltrated, coupled with the scarcity of precipitation [3]. The recommendation for this situation is the construction of surface water capture works and artificial recharge works, since a greater volume of precipitation is expected.

The incidence of extreme temperatures and the scarcity of water is the main cause of concern because crops are lost with frost and more water is lost due to having to replant them. In this sense, it is worth noting that open manifestations regarding the availability of timely and reliable information on the climate and water quality, as well as developing research with a view to other types of crops and/or products, would be of a great benefit.

Climate change accelerates plant extinction by changing their phenology, e.g., mismatching the flowering period of plants with the pollinating time of insects and narrowing the range of physiological adaptation, thus reducing plant resistance to extreme weather events. These effects are particularly substantial in hotspot areas of plant diversity in the tropics and subtropics [16].

Considering that the water used for planting in the REBIVI comes from the extraction of the aquifers, the lack of rain or the higher heat automatically implies a greater use of water. Thus, the elements of greatest impact were the appearance of pests and diseases, drought, increased solar radiation and frost. Interestingly, the presence of rains during the hurricane season (summer–fall) causes concern because they produce erosion and are a vector for diseases. One element to point out is that sprinkler irrigation is in place and workshop participants stressed the need to improve irrigation systems to make a better use of water.

### 4.2. Agriculture

Related to aquaculture, the implementation of reintroduction and repopulation programs of species of commercial interest in the region was proposed, an activity that could be combined with aquaculture through the production of seed and the care of potential recruits until they are released into the marine environment. It should be noted that in the Reserve are examples of very successful activities alternative to fishing, such as gray whale watching in the lagoons, an alternative to the fisheries during the winter season. This point in particular highlights the transversality of the activities proposed in the fishing forums with the topic of tourism within the reserve.

### 4.3. Fisheries

A large number of the communities within the Reserve depend exclusively on fishing activity. They also raised the need for training and financial support to promote them, including the purchase of new equipment specific to each activity, or modifying the existing one to make it more versatile. Some alternatives to the climate change impacts on fisheries are aquaculture and eco-tourism. It is important also to foster the management for sustainable fishing.

### 4.4. Tourism

Undoubtedly, the REBIVI is a special place to promote the endogenous development of the region and tourism industry; where decisions are made about tourism resources based on the capacities and characteristics of the territory, that take advantage of information technologies and promote continuous improvement in institutions. Thus allowing economic growth through sustainable tourism activity in the region, including adaptation actions [43,44].

Examples of EbA actions that are already carried out in the REBIVI and are related to tourism are the special establishments for the conservation of species in danger of extinction, the regulations for gray whale watching, the bans for species of flora and commercial fishing, game hunting, reforestation and land use planning.

Particularly with gray whales, a redistribution of the species has been observed in its breeding and rearing areas under warm El Niño conditions, due to the northward displacement of the optimal temperatures that minimize the thermal stress of newborn calves, with a considerable decrease in the number of animals that use the Bahía Magdalena-Almejas lagoon complex. In this sense, it is expected that an increase in temperature will force a redistribution towards the north of their usual breeding areas, with a potential drop in the number of animals that visit the southernmost lagoons. The significant decrease in the sea ice that covers their feeding areas could be favorable for the region since it would increase the production of offspring and, therefore, the number of animals that visit these lagoons.

*4.5. General*

Top-down initiatives that dominated adaptation efforts globally were focused on hard infrastructure, resulting in bureaucratic and costly measures that failed to improve the long-term adaptive capacity of the poor and the vulnerable. On the other hand, community-based adaptation (CbA) focuses on building local capacity, fostering community participation, integrating indigenous knowledge, prioritizing community empowerment and investing in long-term well-being and resilience [45]. A CbA is a small-scale, place-based and grassroots driven approach that has synergies with broader development aspirations [46]. While A CbA is a recognized approach for enhancing adaptive capacity, gender remains under-represented within the realm of climate change adaptation [47]. The ecosystem-based adaptation (EbA), as one of the most effective mechanisms to deal with the adverse impacts of climate change, has come to be reckoned as a cost-effective measure yielding multiple benefits in the form of goods and services provided by ecosystems [48].

**5. Conclusions**

This paper addresses the local residents' perceptions on defining SEACs, how they are vulnerable to climate change effects and how this is important for the communities' way of life. The results clearly show how biodiversity can lead to community welfare in this NPA, and the importance of climate change adaptation to guarantee the wellbeing of both communities and biodiversity.

The workshops showed an awareness of the problems communities face and some possible solutions to accomplish their goals. In that sense, successful examples of economic diversification as adaptation through a responsible exploitation of biodiversity have been pursued, and, yet, even those positive experiences face important challenges. This means that both human welfare and the conservation of natural resources needs to be placed in the wider context in which economic growth models are put forward. It is also imperative to consider the vulnerability and heterogeneous capabilities of community members in order to better adapt to climate change and strengthen their capacities for action. At the end of the process, the REBIVI's Scientific Subcommittee on Climate Change and Exotic Species validated the actions agreed upon in the seven community workshops.

This included a consideration of issues such as the identification of monitoring strategies, conservation objectives, gender inclusion and equity and women's empowerment. The main lesson was that inclusive mechanisms of education, training and access to better technologies could generate productive and economic conditions. Such changes could lead to greater access to training and the development of skills to improve general standards of living, as well as the empowerment of women through their inclusion in productive work and in making decisions about their own future.

Socio-ecological systems experience constant change and the implementation and monitoring of adaptation actions related to utilization, management, policy, ecological and external influences. As seen, EbA and CbA projects embrace activities that are both community- and ecosystem-focused. Ecosystem conservation and social welfare in the REBIVI would undoubtedly benefit from developing more eco-friendly economic alternatives, increasing knowledge on climate change issues and strengthening institutional and legal capacities to facilitate the implementation and monitoring of adaptation actions. It would be highly relevant to improve regulatory schemes based on responsible public policy, and the participation of experts from the academy.

Finally, it will be important in the future to access research that monitors the progress of the adaptation actions outlined in this study, and how the governance networks in the REBIVI have evolved.

**Supplementary Materials:** The following supporting information can be downloaded at: https://www.mdpi.com/article/10.3390/d14100786/s1, Figure S1. Participants in PCW during the discussion. Figure S2. Process to prioritize the adaptation actions by the communities´ members. Figure S3. Whale watching in REBIVI.

**Author Contributions:** Conceptualization, A.I.B. and P.H.-M.; methodology, A.I.B.; software, P.H.-M.; validation, A.I.B.; formal analysis, A.I.B. and P.H.-M.; investigation, A.I.B. and P.H.-M.; resources, A.I.B.; data curation, A.I.B.; writing—original draft preparation, A.I.B.; writing—review and editing, A.I.B. and P.H.-M.; visualization, A.I.B.; supervision, A.I.B.; project administration, A.I.B.; funding acquisition, A.I.B. All authors have read and agreed to the published version of the manuscript.

**Funding:** This research was funded by the United Nations Program for Development (PNUD) and the National Council for Protected Areas of Mexico (2017–2018).

**Institutional Review Board Statement:** Not applicable.

**Data Availability Statement:** Not applicable.

**Conflicts of Interest:** The authors declare no conflict of interest. The funders had no role in the design of the study; in the collection, analyses, or interpretation of data; in the writing of the manuscript; or in the decision to publish the results.

## Abbreviations

| | |
|---|---|
| CbA | Communities-based Adaptation |
| CCAP | Climate Change Adaptation Program |
| EbA | Ecosystem-based Adaptation |
| IPCC | The Intergovernmental Panel on Climate Change |
| NPA | Natural Protected Area |
| PCW | Participatory Community Workshop |
| REBIVI | El Vizcaino Biosphere Reserve |
| SAA | Strategic Adaptation Axis |
| SAAs | Strategic Adaptation Axes |
| SAA1W | Strategic Adaptation Axis 1: Water |
| SAA2A | Strategic Adaptation Axis 2: Agriculture |
| SAA3F | Strategic Adaptation Axis 3: Fisheries |
| SAA4T | Strategic Adaptation Axis 4: Tourism |
| SEAC | Socio-ecological Asset for Conservation |
| UMA | Unity of Management for Wildlife Conservation |
| UNEP | United Nations Environment Program |

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
