# Peer review of "Impacts of Climate Change in the El Vizcaino Biosphere Reserve (REBIVI): Challenges for Coastal Communities and the Conservation of Biodiversity"

_diversity, doi:10.3390/d14100786_

Round 1
Reviewer 1 Report
This article addresses community perceptions on adaptation to climate change in El Vizcaino Biosphere Reserve (REBIVI), in Baja California Sur (Mexico), one of the largest natural protected areas in Latin America. This manuscript is bad organized and the drawn conclusions are incoherent with the obtained results. The references should be updated to include more recent studies. All the manuscript should be rewritten. It is chaotic and incomprehensible. I have not understood anything. The tables are unreadable and the figures are often to low resolution.
Author Response
The responses to the comments are in the attached table

Reviewer 2 Report
This article addresses a timely and important topic - that of engaging local communities in the adaptation management of nearby protected areas, and species and ecosystems (and ecosystem services) of concern to those communities. I believe that the case study presented could be of interest to a wide readership, but the paper will need to be strengthened in three key areas:
1) Methods - The methods for the case study sound very interesting and relevant; however, the methods are inadequately described to fully convey what was done. For example, for the identification of SEACs, it is unclear how the workshop was designed to translate generic ecosystem services (Table 1) into specific SEACs (Table 2 and Fig 2). The level of detail provided for Stage 1 of the prioritization of adaptation actions (starting line 231) was fairly detailed; but then the Stage 2 methods were unclear and lacked detail. The multi-criterion analysis is an interesting approach, but I am not clear sure how the bulleted list on lines 247-252 all related to climate change adaptation. Other minor suggestions to the methods section include:
* Section 2.3.1 should start with a definition of SEACs - therefore I suggest moving Lines 197-201 to the start of section 2.3.1.
2) Results - The results presented are fairly generic, making them not very informative. For example, the actions listed in Tables 4-7 are vague and it is not clear what exactly is meant by actions, such as "Infrastructure and technological advances" - how are technological advances necessarily climate adaptation actions? The Discussion section ends up provide good illustrative examples of many of these generic action types, but that information belongs in the Results, not in the Discussion section. It is the specifics of what actions were recommended that will be of interest to readers, not overly generic statements. Tables 4-7 also would greatly benefit from doing more to "connect the dots" for readers, about how the actions will address specific climate change impacts, and result in adaptation benefits to the people and SEACs listed. These assumptions are unclear and therefore I don't see how the actions listed are specifically designed to support *adaptation* outcomes. The results section also would benefit from more details on climate change impacts that the actions are designed to address. Some of this information (on climate change impacts, specific actions, and adaptation benefits) are included in the Discussion, but the Discussion is not the proper place for that information. The authors should find a relatively concise way to summarize more of the results in the results section.
3) Discussion - The current discussion focuses more on results, and a disjointed series of points. It should instead focus on synthetic discussion topics that elucidate important key message from the work being presented. The authors need to consider what the top key messages are that they feel their results directly speak to, and focus the Discussion on those points.
Author Response

(The authors gave the same response as above.)

Reviewer 3 Report
Comments:
In the work “Impacts of climate change in the El Vizcaino Biosphere Reserve (REBIVI): challenges for coastal communities and the conservation of biodiversity” (diversity-1823777), the authors addressed community perceptions on adaptation to climate change in REBIVI of Mexico, one of the largest natural protected areas in Latin America, and the data that were derived from workshops with local communities. This is an interesting manuscript and well-written, but some revisions may be needed before acceptance.
Line 15: Too much blank between “analysis.” and “The conclusions”.
Line 25: In the introduction, the adverse effect of climate change on plant diversity loss must be stressed because it may be the fourth factor that reducing plants after habitat loss, deforestation, land use change, but even stronger than any other factors in the mid-century or end of this century or in some parts of the world (doi: 10.1186/s12870-020-02646-3).
Line 48-51: This paragraph is all about the climate, which is very close to the last one, thus it’s better to merge it with the previous paragraph.
Line 54: This study is about how climate change influences the community and biodiversity, when talking about “Changes in climate can cause serious negative consequences”, it’s a good chance to tell the readers what the “serious negative consequences” are. I suggest adding more examples on this point.
Line 81: If you have any questionnaires or survey tables, they can be shown here or as supplementary materials.
Line 84: If this area is an arid and semi-arid area, the drought index, pan evaporation or penitential evapotranspiration and any other eco-hydrological parameters may be shown here.
Line 204: Table 1 is huge. It is more readable if it can be shown as a “round pie chart”, please refer to WWF Living Planet Report 2018 page 18.
Line 280: Figure 2 is very hard to see clearly, a vector-type file is needed to be shown here.
Line 289-302: Tables 4-7 are needed to be remodeled in an article; they are very hard to follow.
Line 303-… In the discussion, I suggest putting more recent work on biodiversity conservation and the sustainability that must be human-centered, such as (1) Gao et al. (2020) Plant extinction excels plant speciation in the Anthropocene. BMC Plant Biology 20: 430. https://doi.org/10.1186/s12870-020-02646-3 and (2) Gao (2021) Applying Humboldt’s holistic perspective in China’s sustainability. Geography and Sustainability 2: 123–126. https://doi.org/10.1016/j.geosus.2021.06.001, put the perspectives and some views in the discussion and conclusion may enhance the readability of this manuscript, e.g., Line 430-431, this assumption can be supported by Gao (2021) even the research areas are different, the core is the same.
Author Response
The responses to the comments are in the attached table.

Round 2
Reviewer 1 Report
The authors addresses community perceptions on adaptation to climate change in El Vizcaino Biosphere Reserve (REBIVI), in Baja California Sur (Mexico), one of the largest natural protected areas in Latin America. This manuscript is now well organized and the drawn conclusions are coherent with the obtained results. However, references should be updated to include more recent studies.
Lines 21 – 23: the key words should be alphabetically arranged
Line 41: It is “Ecosystem-based Adaptation (EbA)”
Line 43: To use only the acronym of REBIVI
Lines 36 – 38: I think that you should add these more recent references to support your sentence “The future does not look promising and it is necessary to develop adaptation policies and instruments in the face of future climatic problems.”. I would like to suggest:
Bosso, L., et al. (2017). Predicting current and future disease outbreaks of Diplodia sapinea shoot blight in Italy: species distribution models as a tool for forest management planning. Forest Ecology and Management, 400, 655-664.
Thornton, P. K., et al. (2018). Is agricultural adaptation to global change in lower-income countries on track to meet the future food production challenge?. Global Environmental Change, 52, 37-48.
Line 77 – 95: In this section, the predictions and hypothesis are not clear. Please, rewrite this part of the manuscript.
Line 122: Please, add the north symbol in the figure.
Lines 139 – 140: I think that you should add these more recent references to support your sentence “as become even more important in the context of global change.”. I would like to suggest:
Bosso, L., et al. (2018). Nature protection areas of Europe are insufficient to preserve the threatened beetle Rosalia alpina (Coleoptera: Cerambycidae): evidence from species distribution models and conservation gap analysis. Ecological Entomology, 43(2), 192-203.
D'Aloia, C.C., et al. (2019). Coupled networks of permanent protected areas and dynamic conservation areas for biodiversity conservation under climate change. Frontiers in Ecology and Evolution, 7, 27.
Line 180: To use only the acronym.
Lines 271 – 272: Please, make uniform (color and style) this table with the other already present in the manuscript.
Lines 374 – 378: Please, move these pictures in the supplementary materials.
Lines 384 – 385: This sections must be called only “Results”.
Lines 392 – 393: All the acronyms should be showed in the extensive form in the table caption.
Reviewer 2 Report
Overall, the authors have done a good job addressing my previous comments. I continued to feel that the work is important to share, and I appreciate the inclusion of more details on the methods so that the work stands a better chance of being replicated by others. I offer a few additional comments below that I think could further strengthen the paper.
Methods – Overall, the authors have done a nice job addressing my comments on the Methods section and providing more details about their methods. I have one further suggestion for adding clarity on the definition of a SEAC:
· Lines 264-266 – “The SEAC is a species/ecosystem that exacerbates its vulnerability to the threats of climate change or, its vulnerability condition is not significantly altered but the ecosystem services it provides to human populations are essential for the livelihoods.” – this wording is a bit awkward and hard to understand. If I am understanding it correctly, I wonder if the following would be more clear: “The SEAC are a species or ecosystem that is vulnerable to the effects of climate change, or a species or ecosystem that provides important ecosystem services to human populations regardless of its vulnerability to climate change.” I also do not understand the reference to gray whales – if the purpose is to provide as example of a SEAC then I suggest the authors include more information about why the gray whale is a SEAC and spell out the example in more detail (which I do think would be helpful to do).
Results – The authors have done a nice job adding more details on adaptation actions to the Results section, and offering more details on how those actions help to support adaptation to specific climate change impacts. I had a few specific suggestions for offering the reader greater clarity on those actions:
· Lines 498-499 – I do not understand what is meant by “hydraulic works”.
· Line 500 – I do not understand what is meant by “sanitary conditions”.
· Line 538 – “The elements of definition and prioritization of adaptation measures in the workshop were…” – the rest of the sentence appears to be commenting on climate change impacts of concern rather than the definition and prioritization of adaptation measures (which I take to be adaptation actions).
· Line 543 – what is meant by “credits”? What type of credits?
· Line 553 – The authors reference “their feasibility” – but it is unclear what is being referred to by “their”.
· Lines 581-587 – It is unclear from this paragraph what the connection is between palm trees and peninsular pronghorn and why, therefore, they are being discussed together.
· For the bulleted lists starting on Line 591 and the second list starting on line 615 – are these meant to relate just to tourism and recreation (since the lists follow Table 7), or are they meant to be more general – i.e., are they meant to integrate across all of the Tables 4-7, and did they result from workshop discussions that were meant to cut across all SSAs?
Discussion – Overall, the Discussion is now more focused on synthetic comments about the project. I would still recommend the following suggestions to further strengthen the clarity and content of this section of the paper:
· Insert subheaders to indicate which of the SSAs are being discussed in each paragraph.
Conclusion – I have two further suggestions regarding the Conclusion:
· Lines 847-849 – As far as I can tell, this is the first and only time the REBIVI Scientific Sub-committee on Climate Change and Exotic Species, and their role in prioritizing adaptation actions, is mentioned. Did that committee shape any of the information that is presented in the Results and Discussion? If so, their role needs to be included in the Methods section and the results that stemmed from their work needs to be clearly flagged in the Results section.
· There is duplicated text in lines 857-866.
Reviewer 3 Report
The authors seem to have done a good job in revising their manuscript. Only one small suggestion before full acceptance, the figures (Lines 374, 376) are hard to see clearly, the authors may need to upload or replace them with high-resolution ones.
Round 3
Reviewer 1 Report
Well done!